# Internal Surface Plasmon Excitation as the Root Cause of Laser-Induced Periodic Plasma Structure and Self-Organized Nanograting Formation in the Volume of Transparent Dielectric

**DOI:** 10.3390/nano10081461

**Published:** 2020-07-26

**Authors:** Vladimir B. Gildenburg, Ivan A. Pavlichenko

**Affiliations:** Faculty of Radiophysics, Lobachevsky State University of Nizhny Novgorod, 603950 Nizhny Novgorod, Russia; iapav@list.ru

**Keywords:** volume nanograting, fused silica, laser pulse, optical discharge, ionization instability, surface plasmon, spatial period

## Abstract

A computer simulation of the dynamics of an optical discharge produced in the volume of a transparent dielectric (fused silica) by a focused femtosecond laser pulse was carried out taking into account the possibility of developing small-scale ionization-field instability. The presence of small foreign inclusions in the fused silica was taken into account with the model of a nanodispersed heterogeneous medium by using Maxwell Garnett formulas. The results of the calculations made it possible to reveal the previously unknown physical mechanism that determines the periodicity of the ordered plasma-field structure that is formed in each single breakdown pulse and is the root cause of the ordered volume nanograting formation in dielectric material exposed to a series of repeated pulses. Two main points are decisive in this mechanism: (i) the formation of a thin overcritical plasma layer at the breakdown wave front counter-propagated to the incident laser pulse and (ii) the excitation of the “internal surface plasmon” at this front, resulting in a rapid amplification of the corresponding spatial harmonic of random seed perturbations in the plasma and formation of a contrast structure with a period equal to the wavelength of the surface plasmon (0.7 of the wavelength in dielectric).

## 1. Introduction

The search for a mechanism predetermining the laser-assisted formation of a regular periodic subwavelength structure inside a solid transparent dielectric continues to be one of the unsolved problems in nanophotonics and laser-matter-interaction physics. The nanogratings formed in the volume of dielectric by fs laser pulses are already widely used in various fields of modern nanoengineering and nonlinear optics. Nevertheless, a clear physical model that reveals the nature of the periodicity of the underlying plasma-field structures formed during the optical breakdown of the dielectric and allows the calculation of their most important characteristic — the spatial period, is still missing. As the numerous experiments have shown, the nanograting in the material of dielectric (predominantly, the fused silica) is oriented perpendicular to the laser pulse polarization, has a period less than the wavelength in dielectric, and can be formed only by the multiple repeated breakdown pulses [1,2,3,4,5,6,7,8,9,10]. Though the intervals between successive pulses (~$10^{-6}-10^{-4}$ s) are much greater than the disappearing time of the plasma structure formed in every single pulse, it is evident that only this structure (at high enough contrast of the its inherent profile of the energy deposition density) can be responsible for the periodicity of the dielectric material properties appearing after a series of repeated pulses.

In the last decade, the main attention in studying these (one-pulse-created) ionization structures was paid to numerical simulation of laser breakdown initiated in fused silica by randomly placed small inclusions with a lowered threshold of ionization (nanobubbles or atom structure defects) playing the role of seed ionization centers. In a number of calculations it was shown that multiple plasmoids formed around these centers during optical discharge evolution and showed a tendency to align in more or less ordered rows forming on average a kind of spatial grating with a period about half to a whole wavelength [11,12,13,14]. These results demonstrate the “plasma-field” nature of the considered structure and the “single-pulse” origin of its periodicity; however, being only based on the direct numerical integration of Maxwell’s equations (with the FDTD method) they cannot reveal the concrete physical nature of periodicity and still do not allow to go beyond some common and evident references to wave interference phenomenon. Alternative models of the formation of nanogratings based on the theory of exciton-polariton interaction [15,16] also encounter difficulties, since they require too low electron density in the discharge, and more importantly, the spatial period in these models is actually an external parameter specified from the outside.

To provide a sufficiently clear physical model that allows us to identify the specific physical mechanism of the emergence of order from chaos in “multiplasmoid” discharge and calculate (at least qualitatively) the main characteristics of the arising ordered structure and their dependencies on external parameters, it seems natural to use approaches based on the concept of ionization-field instability of a medium exposed to high-intensity electromagnetic radiation. The main tasks in these approaches are: (i) description of the joint evolution of the electric field and plasma density perturbations against the background of some quasi-stationary equilibrium states, (ii) obtaining dispersion equations connecting the growth rate constants (so called increments of instabilities) of these perturbations with characteristics of their spatial structure (spatial periods or wave numbers), and (iii) finding the optimal conditions providing enough fast growth of unstable perturbations in a definite range of the wave numbers at the linear and nonlinear stages of instability (corresponding to small and large perturbations, respectively). As applied to the above nanograting problem this program was realized within the frameworks of simple quasi-one-dimensional models of optical discharge maintained in fused silica (in the absence [17,18] or presence [19] of easily ionizable small inclusions) by a traveling plane wave. These models gave results agreeing with experiments in respect of both grating orientation and observed period range of most quickly growing spatial modulation. In particular, one of the ionization-field instabilities (so-called “plasma-resonance”, one caused by quasi-static mutual amplification of the normal electric field component and plasma density within thin under-critical plasma layers) leads to fast growing spatial modulation of plasma and energy deposition density with a maximum of time growth rate lying within the required wave number range (corresponding to the periods from one-third to two-thirds of the laser wavelength in dielectric) [17,19]. A significant drawback of these models from the point of view of explaining the appearance of a regular periodic structure is, nevertheless, the width of this maximum in the space of wave numbers is too large to allow certain spatial harmonics to stand out in the spectrum of random small (seed) perturbations, which would lead, at the nonlinear stage of instability, to the formation of an ordered periodic structure during one breakdown pulse. To form such a structure, the spectrum of eigenwaves of the considered nonlinear dynamical system should apparently contain, along with the given pump wave, another eigenwave, the excitation of which (if its wave number coincides with the wave number of the spatial spectrum of the initial noise) would lead to the appearance of a corresponding sharp resonant maximum in the spectrum of instability increments. In principle, the role of such an additional eigenwave in an optical discharge plasma could be played by a longitudinal Langmuir wave (bulk plasmon), the resonant excitation of which (due to the nonlocality of the polarization response), as it was shown in the work [20], can be significant in optical breakdown of dense gases. However, in the laser plasma parameters range typical of the experimentally observed formation of nanogratings in a solid dielectric, the nonlocal correction to polarizability is negligible compared to its imaginary part and the longitudinal wave is completely suppressed by absorption due to collisions. Therefore, the bulk plasmon cannot be excited in the experiments, and apparently there is no need to take into account the nonlocality of the plasma polarizability (which ensures the possibility of its excitation) in attempts to elucidate the essence of the physical mechanism of nanograting formation.

The difficulty associated with the absence of an additional eigenwave in the volume of a homogeneous medium does not exist in explaining the mechanism of formation of a periodic structure under conditions of optical breakdown at the dielectric–vacuum interface, where the role of such a wave is played by the surface plasmon excited at the interface, the wavelength of which determines the spatial period of the formed surface nanograting [5]. Such surface plasmon can be excited, however, during breakdown inside a homogeneous medium, if a nonuniformity of the parameter determining the ionization rate of the optical field was created in it previously (for example, as a result of the modification of the medium by previous pulses). This is indicated, in particular, by some results of the numerical calculations described above [11,12], according to which an ordered bulk grating was formed at the junction of two half-spaces with different concentrations of seed ionization centers. The possible role of a surface plasmon excitation on the interface between the unperturbed region of dielectric and the region modified by previous pulses is qualitatively considered also in [21].

In the present work, it is shown that the instability of the discharge with respect to the excitation of the “internal” surface plasmon and the periodic plasma field nanostructure generated by it can arise already at the front of the breakdown wave in a single pulse inside a macroscopically homogeneous medium and does not require the creation of a preliminary inhomogeneity of the ionization rate inside it. To describe this phenomenon, the quasi-one-dimensional electrodynamic models, previously used by us, which virtually ignored the role of the longitudinal (parallel to the wave vector of the pump wave) components of the electric field, which are necessarily present in the surface plasmon, were refined, taking into account the excitation of this component on two-dimensional plasma inhomogeneity. As shown by numerical calculations and estimates based on a relatively simple theoretical model that takes into account the generation of a surface plasmon at the breakdown wave front, an ordered subwavelength nanograting is formed in this case in nanodispersed (multyplasmoid) discharge with an arbitrary noise spectrum of small random perturbations. The work is organized as follows. In the second part, the physical model used is described and the basic equations are formulated that describe the spatiotemporal evolution of the discharge in a nanoporous dielectric, which is considered as a continuous heterogeneous medium that can be described on the basis of Maxwell Garnett formulas. The third part presents the results of a numerical calculation that demonstrates the formation of contrast ordered periodic structures of plasma and energy deposition densities, developing from small initial perturbations randomly distributed in space. In the fourth part, a simple model is proposed that makes it possible to qualitatively explain the occurrence of a periodic structure as a result of the excitation of an unstable surface plasmon. In conclusion, the main results of the work are formulated.

## 2. The Physical Model. Initial Equations and Approximations

Our relatively simple (two-dimensional) physical model allows taking into account key electrodynamic factors that determine the possibility of the internal surface plasmon excitation and forming a regular periodic plasma-field structure within a solid dielectric by a single focused laser pulse. The first of these factors is the field amplitude increase in the longitudinal direction inside the beam when approaching the focal plane. It is thanks to this increase that during the ionization process a rather sharp front of the breakdown wave is formed, which can support surface plasmon. The second factor is the relationship between the transverse and longitudinal components of the electric field in the presence of a two-dimensional inhomogeneity of the plasma density. Due to this relationship, the surface plasmon is unstable with respect to arbitrarily weak seed perturbations and imposes its periodicity on the formed structure at the nonlinear stage of instability. Both of these factors are taken into account in our equation:(1)$$\begin{array}{c} \frac{i}{c^{2}}\frac{d(\omega^{2}\varepsilon )}{d\omega }\frac{\partial H_{y}}{\partial t} \end{array}+\frac{1}{S}\frac{\partial }{\partial z}\left(S\frac{\partial H_{y}}{\partial z}\right)+\frac{\partial^{2}H_{y}}{\partial x^{2}}+\frac{\omega^{2}}{c^{2}}\varepsilon H_{y}-\frac{\nabla \varepsilon }{\varepsilon }\nabla H_{y}=0,$$
for the complex amplitude (slow time envelope) of the magnetic field of a quasi-monochromatic TM wave and the equations for the complex amplitudes of the transverse ($E_{x}$) and longitudinal ($E_{z}$) components of the electric field:(2)$$\frac{\partial E_{x}}{\partial t}-i\omega E_{x}=-\frac{1}{\varepsilon }\frac{\partial H_{y}}{\partial z},$$
(3)$$\frac{\partial E_{z}}{\partial t}-i\omega E_{z}=\frac{1}{\varepsilon }\frac{\partial H_{y}}{\partial x}$$
where all the field components are assumed to be written as real parts of the productions of their complex amplitudes and factor exp(−iωt). In Equations (1)–(3), ε is a dielectric function of the ionized medium. When using Equation (1) for a slow field envelope, it is assumed that the characteristic time of the change in the complex field amplitude and plasma density is large compared to the field period 2π/ω. The function S(z) in the second term of Equation (1) can be considered here as the effective ray tube cross-section in the near-axis region of the focused wave beam, the field of which is modeled by the above equations. Equation (1) can be obtained from the exact wave equation based on the following approximations: (i) we neglect the second derivative of the field amplitude with respect to time; (ii) we replace the Laplace operator with the mentioned term, which allows us to describe the axial changes in the field amplitude of the focused beam in its axial region under the assumption of a given (converging) ray tube and the field amplitude being constant on the wavefront surface orthogonal to the rays. Similar approximations in the study of the breakdown wave dynamics in a paraxial wave beam were used earlier, for example, in [22]. Modifying the wave equation in this way, we get the opportunity to describe the (important to us) longitudinal changes in amplitude due to the focusing of the beam, at the same time distracting from its large-scale transverse inhomogeneity. Consideration of the latter would be important only in calculating the total transverse size of the emerging structure and is not directly related to the main issue that interests us here, concerning the nature of the small-scale transverse periodicity of the field. In fact, ignoring the large-scale transverse inhomogeneity in the framework of this model, we attribute our study to the structure created in the wide axial region of the long-focus beam. In the calculations below, we take for the ray tube cross-section an expression $S=1+z^{2}/l_{F}^{2}$ corresponding to an unperturbed Gaussian beam with a Rayleigh length $l_{F}$ and a known axial distribution of the field:(4)$$E_{x}^{\left(0\right)}=\frac{H_{y}^{\left(0\right)}}{\sqrt{\varepsilon_{s}}}=E_{F}{\left(1+\frac{z^{2}}{l_{F}^{2}}\right)}^{-1/2}\exp\left(ik_{s}z-\frac{{\left(t-z\sqrt{\varepsilon_{s}}/c\right)}^{2}}{\tau^{2}}\right),\;k_{s}=\frac{\omega }{c}\sqrt{\varepsilon_{s}},$$
obtained in this case for a Gaussian pulse (with a duration $\tau_{p}$ and maximum electric field amplitude $E_{F}$) directly based on the solution of Equations (1)–(3) in the absence of ionization (i.e., in the homogenous dielectric with permittivity $\varepsilon_{s}$). Additional formal justifications for the possibility of using the simplified wave Equation (1) are given in Appendix A.

To study the joint evolution of the field and plasma in the discharge produced by such a pulse, which is our main task here, a numerical solution of Equations (1)–(3) is required together with equations describing the kinetics of the discharge and dielectric function of the ionized medium. In the case of apparently the most practical interest, we should consider the discharge arising in the dielectric (fused silica) as a result of the ionization of multiple foreign inclusions with a reduced ionization threshold that can play the role of primary centers of breakdown. Generally speaking, a fairly complete description of the dynamics of such a discharge is a very complex and time-consuming task that requires taking into account the spatiotemporal evolution of many inhomogeneous plasmoids formed near each of primary ionization centers, describing the processes of their expansion, shape change, and merging (see, for example, [11,12,13,14]). However, the essence of the physical mechanism that determines the periodicity of arising structures, as it seems to us, can be revealed in the framework of the simplified model that we use. We assume that ionization occurs in the volume of many small spheres with a lowered ionization threshold U=5.2 eV [12,14,23], the sizes and number of which are fixed. Within this approach, a microscopically inhomogeneous medium can be considered as a continuous heterogeneous one; the fields in the Equations (1)–(3) should be considered as their macroscopic means; and the dielectric function ε should be replaced by its effective value $\varepsilon_{eff}$, which is determined based on the well-known Maxwell Garnett formulas:(5)$$\varepsilon_{eff}=\varepsilon_{s}\frac{1+2f\alpha }{1-f\alpha },\;\alpha =\frac{\varepsilon_{p}-\varepsilon_{s}}{\varepsilon +2\varepsilon_{s}},$$
(6)$$\varepsilon_{p}=\varepsilon_{s}-\frac{N}{N_{c}}\left(1-i\nu /\omega \right),$$
where $\varepsilon_{p}$ is permittivity of the plasma, N is free electron density, $N_{c}=m(\omega^{2}+\nu^{2})/(4\pi e^{2})$ is its critical value, ν is the effective frequency of electron collisions, and f is the volume fraction of spherical plasmoids. Figure 1 shows the dependences of the real and imaginary parts of the effective permittivity $\varepsilon_{eff}$ on the dimensionless plasma density $n=N/(\varepsilon_{s}N_{c})$ at the values ν/ω=0.15 (which seems to us the most realistic for the conditions for producing nanogratings in fused silica based on the results of [23,24,25]) and volume fraction f=0.3 (close to that which can be extracted from references [11,12,13,14]).

The change in plasma density is determined by the rate equation
(7)$$\frac{\partial N}{\partial t}=W_{pi}+W_{a}-\frac{N}{\tau_{r}}+W_{n}.$$
The first three terms on the right-hand side of this equation describe multiphoton and avalanche ionization, and recombination. We have used the approximations for dependences of their rates on laser intensity similar to the ones used in the previous studies [23,24,25] of the near-infrared (800 nm) breakdown of the nanoporous fused silica with lowered bandgap U=5.2 eV:(8)$$W_{pi}=\sigma_{3}(N_{m}-N){\left({\left|\eta \right|}^{2}I\right)}^{3},\;W_{a}=\frac{8\pi \nu e^{2}}{3\sqrt{\varepsilon_{s}}mcU\omega^{2}}\frac{N_{m}-N}{N_{m}}{\left|\eta \right|}^{2}IN,$$
$$\eta ={\left[1+(1-f)(\varepsilon_{p}-\varepsilon_{s})/(3\varepsilon_{s})\right]}^{-1}.$$
Here $I=c\sqrt{\varepsilon_{s}}{\left|E\right|}^{2}/8\pi$ is the laser intensity, η is the proportionality coefficient between the field inside the sphere and the averaged macroscopic field, $\sigma_{3}=1.5\times 10^{-28}$ W^−3^cm^6^s^−1^, $N_{m}=2.1\times 10^{22}{\;\mathrm{cm}}^{-3}$ is the atom density, and $\tau_{r}=150$ fs [26] is the characteristic recombination time. The last term $W_{n}$ is a “noise” ionization source introduced into the equation for modeling small seed fluctuations of the plasma density at the initial stage of breakdown in the main region of the discharge ($|z|<l_{F}$). It should be noted that the introduction of this source is only a formal technique, allowing us to take into account small initial (seed) fluctuations caused by the statistical spread (even if very small) of the parameters of discrete inclusions in the real discharge. If we try to give this source a more specific and clear meaning and talk about the actual origin of the fluctuations, we can, for example, consider it as a fluctuating component of the ionization rate due to the inevitable presence of small noise fluctuations in the field amplitude in the incident laser pulse.

The equations describing the spatiotemporal evolution of the field and plasma were numerically solved in a rectangular region $0\leq x\leq L_{x}$, $L_{1}\leq z\leq L_{2}$. For simplicity, the boundary conditions along the transverse coordinate x were assumed to be periodic with a common period of several wavelengths ($2<L_{x}/\lambda <15$) in the unperturbed dielectric and significantly exceeding the periods of spatial harmonics of the field and plasma perturbations of interest to us. The boundaries along the longitudinal coordinate z were chosen so that the computational domain includes with sufficient margin the entire region ($|z|<l_{F}$) in which the plasma produced can affect the field. In order to get rid of unwanted reflection of waves from the z-boundaries of the computational domain, absorbing layers were placed at these boundaries, providing a sufficiently low reflection coefficient (~10^−5^). A noise ionization source generating a random spatial distribution of small initial density fluctuations was specified as a random set of sufficiently large number (10×10=100) of sinusoidal harmonics:(9)$$\begin{array}{l} W_{n}=A\cos^{2}\left(\frac{\pi }{2l_{F}}z\right)\sum_{m=1}^{10}\sum_{n=1}^{10}\cos\left[\frac{\pi m}{L_{x}}(x+\alpha_{mn}z)+\phi_{mn}\right]\;\mathrm{at}\;|z|<l_{F},\;t_{1}<t<t_{2},\\ W_{n}=0\;\mathrm{at}\;|z|\geq l_{F},\;t<t_{1},\;t>t_{2}, \end{array}$$
where $\phi_{nm}$ and $\alpha_{nm}$ are random variables (with equal probability distributed over the segments [0,2π] and [−1,+1], respectively) that determine the random phases and directions of the wave vectors of spatial harmonics of perturbations. The amplitude A and times of the beginning ($t_{1}$) and end ($t_{2}$) of the action of the noise source were set so that by the time when the background (unperturbed) plasma density reached a predetermined small value $N=0.01N_{c}\varepsilon_{s}$, the maximum value of the density fluctuations was 0.01 of this value.

Along with the plasma density and field, we also calculated the spatial distribution of the local energy deposition density in the medium w(x,z,t) by the given time t (see also [27]):(10)$$w=f\int_{-\infty }^{t}\left[\nu \frac{N}{N_{c}}{\left|\eta E\right|}^{2}+\left(8\pi U+\frac{1}{2}\frac{{\left|\eta E\right|}^{2}}{N_{c}}\right)s\frac{\partial N}{\partial t}\right]dt.$$
Here, s=1 at ∂N/∂t>0 and s=0 at ∂N/∂t<0; the first term under integral describes the electron collision losses, the second one takes into account the energy expenditure for the interband transition (overcoming the bandgap U) and energy transfer to the newly born free electrons. This function is an important characteristic determining the rate and space profiles of the medium heating and therefore the course of the thermo-mechanical and chemical processes and accumulation effects [14,28] responsible for the volume nanograting formation in the medium by the repeated pulses. The high contrast nanograting can be formed evidently if by the end of the pulse the sufficient energy deposition w is localized in a comparatively narrow regions.

## 3. Results of Calculation

Numerical simulation was performed at laser pulse parameters typical of the experiments on the formation of nanogratings in the volume of fused silica: the maximum value of the unperturbed intensity at the focus $I_{\max}=c\sqrt{\varepsilon_{s}}|E_{F}{|}^{2}/(8\pi )$~$10^{13}-10^{14}\;{W/\mathrm{cm}}^{2}$, the dimensionless focal length $k_{s}l_{F}=35$ (beam convergence angle $\vartheta =10^{{}^{\circ}}$), and the pulse duration τ=100 fs. Initially, the discharge dynamics were simulated in the absence of a noise ionization source that produces small seed fluctuations in the plasma density N in spherical plasmoids and the effective dielectric function $\varepsilon_{eff}$ determined by it. In this case, the general scenario for the development of the discharge is characterized by the formation of a one-dimensional breakdown wave, which arises in the focus region and counter-propagates to the incident beam. The nature of the evolution of the longitudinal profile $N_{0}(z,t)$ of this wave and the maximum density value $N_{0\max}$ achieved at its front depends on the beam intensity $I_{\max}$ and changes significantly when passing through some critical values $I_{1}\approx 2\times 10^{13}$ W/cm^2^ and $I_{2}\approx 7.3\times 10^{13}$ W/cm^2^. Figure 2 shows the spatiotemporal evolution of dimensionless density profiles $n_{0}=N_{0}(z,t)/(\varepsilon_{s}N_{c})$ at three intensities. At low intensities $I_{\max}<I_{1}$ (Figure 2a), i.e., at a relatively low ionization rate, the plasma density in spherical plasmoids does not have time to reach a threshold value $N_{th}$ corresponding to the condition Reεeff=0 during the pulse width. In the region $I_{1}<I_{\max}<I_{2}$ (Figure 2b), a thin (compared to wavelength) layer with a suprathreshold density ($N_{0\max}>N_{th}$) is formed, the profile of which remains almost unchanged for quite a long time (actually from the moment the density passes through this threshold until the end of the ionization process). 

A further increase in intensity $I_{\max}>I_{2}$ (Figure 2c) leads again to an expansion of the density profile and a decrease in its maximum because of the formation of an extended but not sufficiently ionized region in the front (far extending toward the beam) part of the discharge. The absorption of the incident wave in this region, due to its large extent, leads to a strong decrease in the field and the actual stop of the breakdown process at a relatively low level of plasma density already in far approaches to the focal region. 

The main characteristics of the suprathreshold plasma layer formed in the discharge as functions of the intensity are illustrated in Figure 3, which shows the intensity dependences of the dimensionless maximum density,$n_{0\max}=N_{0\max}/(\varepsilon_{s}N_{c})$, maximum suprathreshold layer thickness $L_{\max}$, (Figure 3a), and its lifetime $T_{0}$, (Figure 3b). As can be seen from the figure, in the entire indicated range of intensities, the formed layer is characterized by a relatively small thickness (shorter than the wavelength) and a sufficiently long lifetime (~60–80 fs), which contributes to the excitation of an internal surface plasmon on this layer and to the formation of a periodic structure (see below).

The spatiotemporal evolution of the discharge in the presence of small seed noise (Figure 4), which makes it possible for the instability we are interested in, is illustrated in Figure 5, Figure 6 and Figure 7 for the intensity value $I_{\max}=4\times 10^{13}$ W/cm^2^ lying in the intermediate region of greatest interest $\left(I_{1},I_{2}\right)$, where the peak density at the front of the unperturbed breakdown wave goes over a threshold value $N_{th}$. It is after such a transition (Figure 5) that transverse density modulation (with a period approximately equal to 0.67λ) arises and rapidly increases on this front, leading to the formation of a contrasting periodic structure. We note that randomly placed plasmoids are not shown in our figures. Their coordinates do not appear in our consideration; their sizes are much smaller than the characteristic scale of the shown density distributions. The fact that there is a close correlation between the effects of the formation of a thin layer of above-threshold density (with $N_{0\max}>N_{th}$, i.e., Reεeff<0) and its strong transverse modulation, together with the results of a qualitative analysis based on the simple model presented below, allows us to conclude that the excitation of the internal surface plasmon is crucial.

The appearance of this plasmon as a new natural wave in the system under study, as shown in Section 4, leads to a resonant increase in the instability increment of the spatial harmonic of perturbations synchronous with it (whose wave vector coincides with the surface plasmon wave vector). The released harmonic relatively quickly leads to a strong corrugation of the breakdown wave front with a period equal to plasmon wave length; due to the strong modulation of the field amplitude (and hence the ionization rate) along the front, the discharge decays into separate layers with suprathreshold density (Figure 6 and Figure 7). At the final stage of the process, these layers (in fact the same as in the case of an initially purely harmonic perturbation with a given period [18,27]) lead to the formation of a contrast grating structure characterized by a strong nonuniformity in the transverse direction distribution of the energy deposition density (Figure 8). Within the parameter range used in calculations, the spatial period of this structure Λ turned out to be close to $0.67\lambda_{s}\approx 350$ nm, which fits into the framework determined by the experimental data. It is also important to note that, as the simulation showed, this quantity (in the interval of parameters where an ordered structure appeared) did not depend on the concrete realization of the random initial distribution of small perturbations and the sizes of the calculation region. However, there are some factors that are not accounted for in our model but can affect the plasmon wavelength and, therefore, the period of the structure. Probably the non-sphericity of the small plasmoids formed under real conditions in the nanodispersed discharge belongs to them in the first place. This factor can be analyzed within the framework of a more advanced model (based on the same approaches as the model we proposed), in which the discharge in fused silica is described as anisotropic nanodispersed media, whose permittivity tensor components are calculated on the basis of extended Maxwell Garnett theory. Also, the plasmon wavelength can be influenced by the curvature of the breakdown wave front, which arises due to the smooth transverse inhomogeneity of the laser beam field and by the inhomogeneity of the ionized medium due to its heating by a series of multiple laser pulses. Taking these factors into account, however, significantly complicates the model and goes beyond the scope of our work, the purpose of which is to clearly demonstrate, based on the simple model, the physical mechanism determining the periodicity of the nanogratings.

Outside the intensity interval $\left(I_{1},I_{2}\right)$ (i.e., at $N_{0\max}<N_{th}$) it can lead, during the evolution of the discharge, to the formation of small-scale high-density regions, but the regular periodic structure of interest to us does not form here.

## 4. Instability of a Thin Plasma Layer with Respect to the Excitation of a Surface Plasmon

As a structure simulating an unperturbed plasma density profile at the moment of formation of a narrow density peak in the breakdown wave (Figure 2b), we consider a stationary discharge supported in the form of a thin uniform plasma layer by a given external field $x_{0}E_{0}\exp(-i\omega t)$ parallel to its boundaries z=±l (see Figure 9a); the stationary plasma density in the layer is $N_{0}$. Its dielectric constant $\varepsilon_{0}$, in view of the relatively small value of the parameter ν/ω=0.15, used in our calculations, is assumed to be purely real in the simplified model under consideration. As is known, various types of surface waves can propagate along the plasma layer. Among them, the main interest for us is the “antisymmetric” surface plasmon, in which the field component $E_{x}\sim \exp\{-i(\omega t-hx)\}$ is an odd function of the coordinate z perpendicular to the layer. As follows from the dispersion equation:(11)$$D(h)\equiv \tanh\kappa_{0}l+\varepsilon_{0}\frac{\kappa_{e}}{\kappa_{0}}=0,\;\kappa_{0}=\sqrt{h^{2}-k_{0}^{2}\varepsilon_{0}},\;\kappa_{e}=\sqrt{h^{2}-k_{0}^{2}},\;k_{0}=\frac{\omega }{c},$$
(see, for example, [29]), for a small layer thickness ($k_{0}l<<1$) this plasmon can exist even if the stationary plasma density $N_{0}$ slightly exceeds a threshold value $N_{th}$ (i.e., for $-\varepsilon_{0}<<1$). Consequently, as the plasma density in the layer increases from zero, this plasmon begins to propagate first (see Figure 9b), and at the threshold of occurrence ($\varepsilon_{0}\approx -2k_{0}l$) it has a wavelength $\lambda_{0}=2\pi /h_{0}=\lambda /\sqrt{2}$, regardless of the specific values of the layer parameters.

We now turn to an analysis of the instability of the discharge in the layer with respect to perturbations of the plasma density and the field amplitude of the form $N_{1},E_{1}\sim \exp(\gamma t)\cos(qx)$ and write down the equations relating these quantities in the plasma in the linear approximation [17,19]
(12)$$\begin{array}{c} \Delta E_{1x}+k_{0}^{2}\varepsilon_{0}E_{1x}-E_{0}k_{0}^{2}\frac{N_{1}}{N_{c}}-E_{0}\frac{\partial^{2}}{\partial x^{2}}\frac{N_{1}}{N_{c}}=0,\\ \frac{\partial N_{1}}{\partial t}=\beta \frac{E_{1}}{E_{0}}-\alpha N_{1},\;\alpha =N_{c}\frac{\partial W(E_{0},N_{0})}{\partial N_{0}},\;\beta =\frac{E_{0}}{N_{c}}\frac{\partial W(E_{0},N_{0})}{\partial N_{0}}, \end{array}$$
where W is the total ionization rate. For a given dependence of perturbations on x, Equation (12) (taking into account the boundary conditions for the continuity of the tangential components of the electric and magnetic fields at the layer boundaries) leads to the following problem of finding the eigenfunctions $E_{1x}(z)$ and eigenvalues $p_{i}$:(13)$$\begin{array}{c} \frac{\partial^{2}}{\partial z^{2}}E_{1x}-p_{i}^{2}E_{1x}=0,\\ {\left(\frac{\partial E_{1x}}{\partial z}\pm \frac{p_{0}^{2}}{\varepsilon p_{e}}E_{1x}\right)|}_{z=\pm l}=0,\\ p_{0}^{2}=q^{2}-k_{0}^{2}\varepsilon ,\;p_{e}^{2}=q^{2}-k_{0}^{2},\\ p_{i}^{2}=p_{0}^{2}+\left(1-\frac{q^{2}}{\varepsilon_{0}}\right)\frac{\beta }{\gamma -\alpha }. \end{array}$$

The indicated problem has two types of solutions (symmetric and antisymmetric in the z coordinate), the eigenvalues of which $q_{i}$ are found as the roots of the equation
(14)$$\tanh p_{i}l=-{\left(\frac{\varepsilon p_{e}p_{i}}{p_{0}^{2}}\right)}^{\pm 1},$$
(plus and minus correspond to symmetric and antisymmetric mods), determining (based on the last of the Equation (13)) time increments of perturbations as a function of their wave numbers q (or periods Λ=2π/q); in particular for antisymmetric perturbation
(15)$$\gamma (q)=\alpha +\frac{l^{3}{\left(q^{2}-k_{0}^{2}\varepsilon \right)}^{3/2}}{3\varepsilon_{0}}\frac{\beta }{D(q)}.$$

An analysis of Equation (14) shows that, at low concentrations ($\varepsilon_{0}>0$), the increments of both types of perturbations are finite, however, with increasing concentration in the layer, the possibility of excitation of a surface plasmon appears, as a result of which the instability increment $\gamma (q,\varepsilon_{0})$ becomes infinitely large for an antisymmetric perturbation with a wave number equal to the wave number of the surface plasmon $q=h_{0}$. Under realistic conditions (at a finite frequency of collisions) this infinity is obviously absent, however, as estimates made on the basis of Equation (15) show, the perturbation increment with a period equal to the plasmon wavelength $\gamma (h_{0},\varepsilon_{0})\sim \beta \omega /\nu$ significantly exceeds the increments of all other perturbations, this harmonic quickly goes into the nonlinear stage and after its standing out determines the further evolution of the discharge as a whole. Thus, the excitation of a surface plasmon is apparently the main factor ensuring the separation of a single harmonic of the perturbation and leads to the appearance of an ordered periodic structure from initially chaotic perturbations.

## 5. Conclusions

We have suggested the physical model that explains the laser-pulse-induced nanograting formation within the volume of a solid dielectric as a result of ionization-field instability of the optical discharge in respect to excitation of the “internal surface plasmon” generated at the narrow leading edge of the breakdown wave counter-propagated to the incident pulse inside the dielectric. The instability we consider, like other ionization-field instabilities, is due to the effects of mutual amplification of small perturbations in the field amplitude and the plasma density. The mechanism of its occurrence can be briefly explained as follows. At the leading edge of the breakdown wave counter-propagating with the incident beam, a thin plasma layer is formed in which the effective permittivity is negative. A surface wave (surface plasmon) with a wavelength shorter than the incident wavelength can propagate along such a layer. This plasmon is not excited by the incident electromagnetic wave as long as the layer remains uniform in the longitudinal direction. However, with the appearance of an infinitesimal periodic perturbation of the plasma density along the layer with a spatial period equal to the wavelength of the plasmon itself, it is effectively excited (in the form of a standing surface wave). The maxima of the amplitude of the total electric field (and, consequently, the maxima of the ionization rate) arising from the interference between the fields of the plasmon and the incident wave, that excited it, are located in the same places where the maxima of the density perturbation are located. Due to an increase in the ionization rate in these places, the density maxima will continue to grow, and thus, a positive feedback loop arises, indicating the appearance of instability. The role of the initial (seed) perturbation is played by the corresponding spectral component, which is always present in the continuous spectrum of weak random fluctuations of parameters (in our case, plasma density), which are always present in any real system. We have carried out numerical simulation based on the equation system including the rate equation for the plasma density and wave equation describing the spatiotemporal evolution of the average field in nanoporous fused silica that we have considered as a heterogeneous medium with effective dielectric permittivity calculated using the Maxwell Garnett formulas. We have found that a thin plasma layer with negative (and close to zero) effective permittivity was formed in the head part of the breakdown wave counter-propagated to the incident laser pulse. Surface plasmon have excited at this layer resulting in strong modulation of plasma density along it (in the direction of laser polarization), strong corrugation of the breakdown wave front and formation of contrast periodic structure that can be considered probably as a root cause of subwavelength nanograting formation in the dielectric material. The spatial period of the formed periodical nanostructure (coinciding according to our qualitative model with the wave length of the internal surface plasmon) did not depend on the structure of random small initial perturbations playing the role of a small seed which is necessary to any instability. Within the parameter range used in our calculations this period was about 0.7 of the laser wavelength in unperturbed dielectric which fits within the framework of experimentally measured values. The experimentally observed dependences of the nanograting period on laser pulses parameters could be explained probably only based on a more complicated model (in particular, out of Maxwell Garnett approximations and with thermo-chemical processes between pulses taken into account).

## Figures and Tables

**Figure 1 nanomaterials-10-01461-f001:**
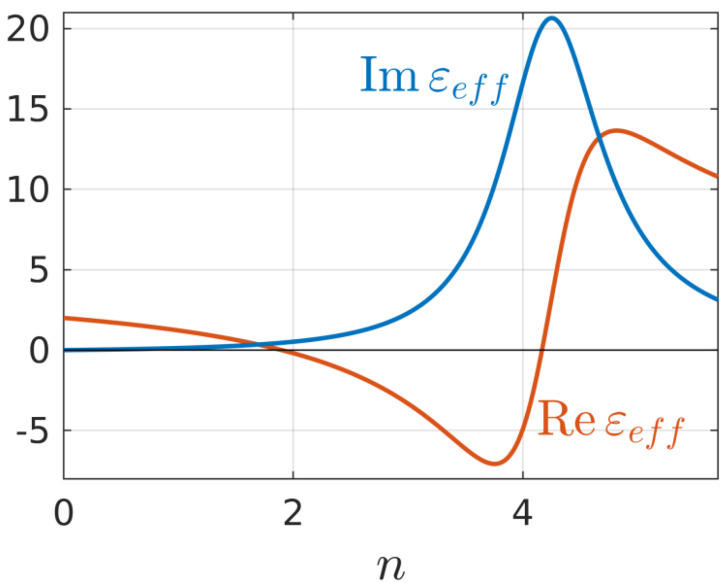
Dependences of real and imaginary parts of the effective medium permittivity on dimensionless density $n=N/(\varepsilon_{s}N_{c})$ inside the plasmoids at volume fraction f=0.3, and effective electron collision frequency ν/ω=0.15.

**Figure 2 nanomaterials-10-01461-f002:**
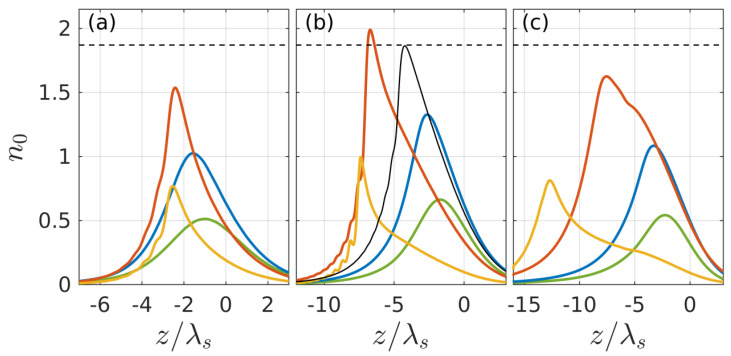
Snapshots of plasma density in an unperturbed breakdown wave $n_{0}$ (in the absence of seed noise) at different instants of time for different values of the incident pulse intensity (**a**) $I_{\max}=1.5\times 10^{13}$ W/cm^2^ (**b**) $I_{\max}=4\times 10^{13}$ W/cm^2^, (**c**) $I_{\max}=10^{14}$ W/cm^2^. The green, blue, red, and yellow curves correspond to successive instants of time (**a**) t=0, 19, 53, 112 fs (**b**) t=−32, −24, 18, 98 fs, (**c**) t=−56, −50, −32, 75 fs. Red curves correspond to maximum density values $N_{0\max}$. The black curve (t=−12 fs) in panel (**b**) shows the plasma density profile at the moment of passing through the threshold value $N_{th}$ corresponding to Reεeff=0 (dashed line). The laser pulse propagates in the z direction from left to right; its maximum reaches the focus (*z* = 0) at *t* = 0. Effective medium and laser pulse parameters are the following: volume fraction of foreign inclusions f=0.3, effective collision frequency ν/ω=0.15, laser pulse duration τ=100 fs, dimensionless focal length $k_{s}l_{F}=35$ (convergence angle $10^{\circ }$), wavelength $\lambda_{s}=2\pi /k_{s}=560\;\mathrm{nm}.$

**Figure 3 nanomaterials-10-01461-f003:**
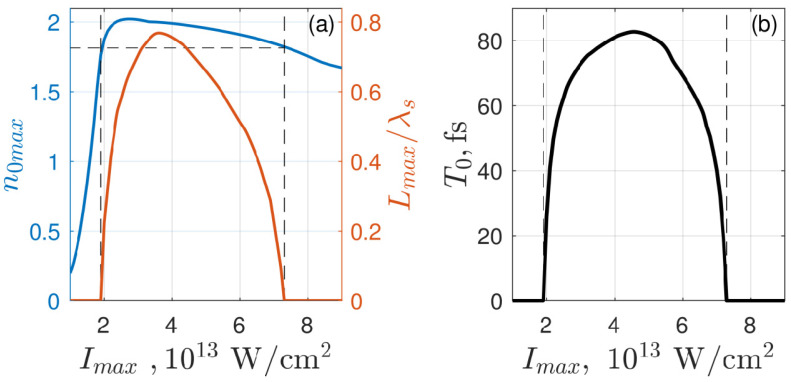
Dependences on the laser pulse intensity of (**a**) maximum values of the dimensionless plasma density $n_{0\max}$ (blue curve) and width of suprathreshold plasma layer $L_{\max}$ (red curve) and (**b**) lifetime of this layer $T_{0}$. Horizontal and vertical dashed lines indicate the threshold plasma density value and boundaries of the range of intensities in which this threshold is overcome, respectively. Effective medium and laser pulse parameters are the same as in Figure 2.

**Figure 4 nanomaterials-10-01461-f004:**
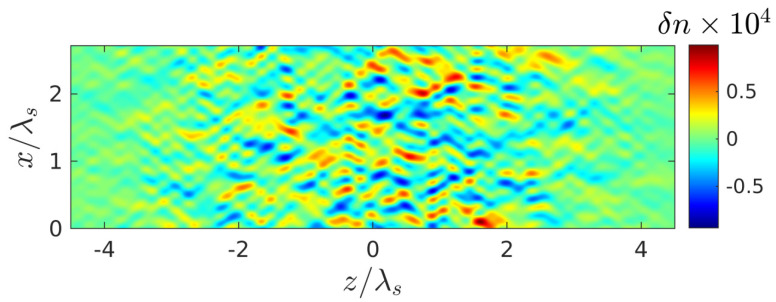
Spatial distribution of seed perturbations of plasma density in plasmoids $\delta n=n-n_{0}$ by the time the noise ionization source expires $t=t_{2}=-83$ fs; the laser pulse and effective medium parameters on current and further figures of this part of the work are the same as in Figure 2b.

**Figure 5 nanomaterials-10-01461-f005:**
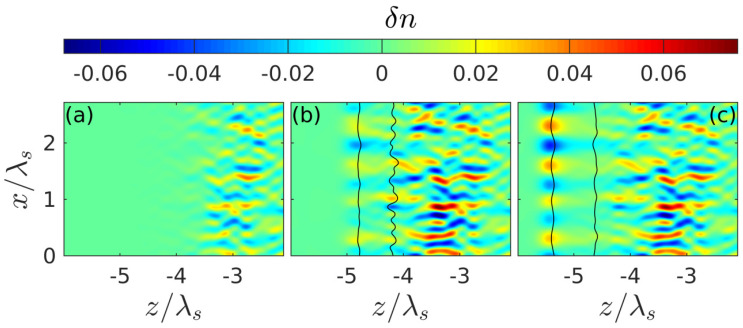
Density perturbation δn snapshots illustrating formation of a regular periodic structure during the transition of the density $n_{0}$ through the threshold value (black curve in Figure 2b) at the intensity $I_{\max}=4\times 10^{13}$ W/cm^2^ lying in the optimal range. Panels (**a**–**c**) correspond to time instances t=−19 fs, t=−12 fs and t=−5 fs, respectively. Black curves show the boundaries of the region where the plasma density exceeds the threshold value (Reεeff<0).

**Figure 6 nanomaterials-10-01461-f006:**
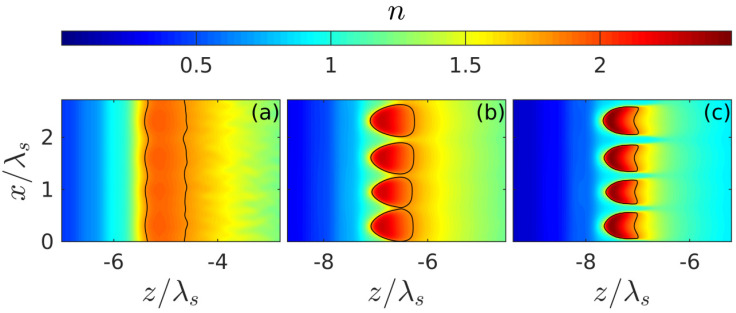
Formation of the plasma grating after the optimal scale of the instability has stood out. Plasma density distributions n(x,z) are shown at time instances (**a**) t=−5 fs (**b**) t=18 fs and (**c**) t=45 fs; the intensity $I_{\max}=4\times 10^{13}$ W/cm^2^.

**Figure 7 nanomaterials-10-01461-f007:**
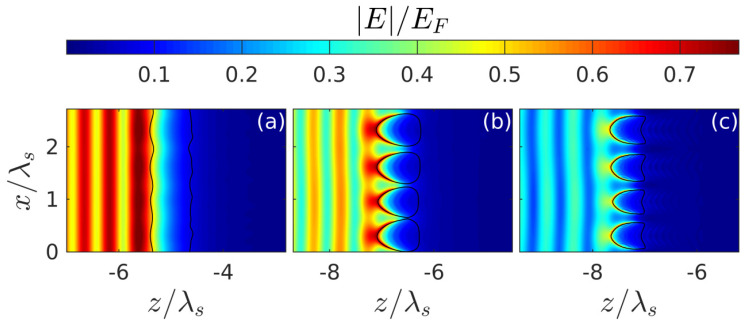
Shapshots of the effective field amplitude at same time instances as for Figure 6; the maximum electric field amplitude in the incident pulse; $E_{F}=1.5\times 10^{8}$ V/cm ($I_{\max}=4\times 10^{13}$ W/cm^2^).

**Figure 8 nanomaterials-10-01461-f008:**
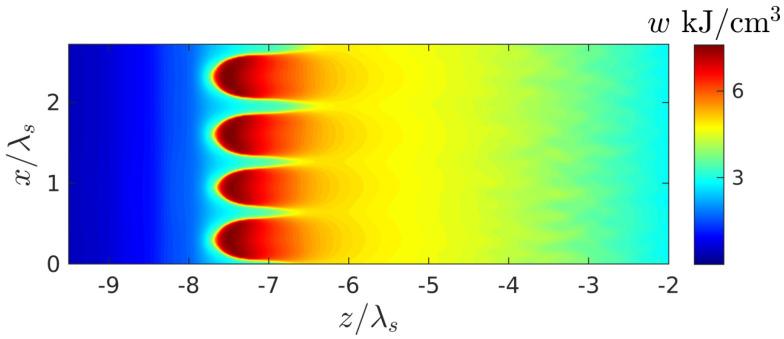
Spatial distribution of the resulting energy deposition density w(x,z) at the end of the breakdown pulse at the intensity $I_{\max}=4\times 10^{13}$ W/cm^2^.

**Figure 9 nanomaterials-10-01461-f009:**
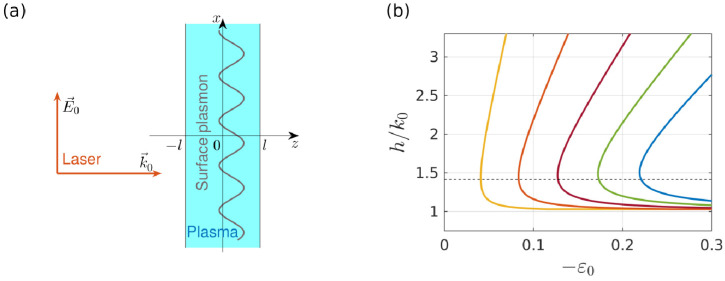
(**a**) The geometry of the model problem. (**b**) Dependences of the surface plasmon wave number h on permittivity of the thin plasma layer at its different widths within our qualitative model. Curves correspond (from left to right) to the values of the parameter $k_{0}l$ increasing from 0.02 to 0.1 with a step 0.02. The dashed line shows the plasmon wave number at its appearance threshold ($h_{0}=\sqrt{2}k_{0}$ at $\varepsilon_{0}\approx -2k_{0}l$).

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
