# Peer review of "Internal Surface Plasmon Excitation as the Root Cause of Laser-Induced Periodic Plasma Structure and Self-Organized Nanograting Formation in the Volume of Transparent Dielectric"

_nanomaterials, 2020, doi:10.3390/nano10081461_

Round 1

Reviewer 1 Report

I think it is an excellent paper dealing with the long standing problem of formation of periodic nanostructure formation during intense fs pulses.

Explanation and results of the model sounds logical to me. I have 1-2 remarks to the authors.

In the introduction the authors indicate that the Results of Ref. [15,16 ] require the lower density in the discharge. Can they benchmark the results of the model to those conditions ? Do not they encounter the similar problem ?

In Figure 8 the spatial distribution of perturbation ~ 2/3 lambda... There were the observations as authors cite in Introduction with 1/3 lambda... 

What are the crucial parameters to achieve the periodicity of 1/3 lambda ?

Best Regards,

Author Response

Dear Reviewer,

We thank you for carefully reading the article and your comments. We answered your questions (see the attached file), took into account your comments and made the appropriate changes and additions to the article.

Sincerely,
Authors

Reviewer 2 Report

This paper uses a reduced model to describe nanograting formation in ultra-short pulse laser interaction with dielectric.  The model is made more tractable by treating small scale plasma sites as an effective medium.  The results are very promising.

I have two major requests.  First please derive equation 1 at least in outline, or perhaps put a full derivation in an appendix, or provide a citation where it is clearly derived.  Second I could not follow the geometry in section 4, please provide a figure illustrating the orientations and boundary conditions.  Why is the system bounded as -l<z

Author Response

Dear Reviewer,

We thank you for carefully reading the article and your comments. We answered your questions (see the attached file), took into account your comments and made the appropriate changes and additions to the article.

Sincerely,
Authors

This manuscript is a resubmission of an earlier submission. The following is a list of the peer review reports and author responses from that submission.

Round 1

Reviewer 1 Report

This paper introduces a method of simulating optical discharges caused by focused femtosecond laser pulses in a dielectric medium. With the help of Maxwell Garnett formulas as well as the consideration of small-scale ionization-field instability, the authors developed a model which could calculate the periodicity of the plasma-field structure in the dielectric material. The idea of thin plasma layer and the so-called “internal surface plasmon” are the key points in this model, which leads to some calculation results matching existing experimental data. Therefore I would recommend it to be published in Nanomaterials. Some comments on this work are:

  1. In page 6, the authors mentioned two critical values of 2x1013 W/cm2 and 9x1013 W/cm2. Please provide more explanations as well as necessary calculations in order to get those two critical values.
  2. In page 7, it is claimed that “a thin layer with a suprathreshold density is formed, the profile of which remains unchanged for quite a long time”. Please specify why that profile is “unchanged”. Moreover, the “suprathreshold-density layer thickness v.s. laser beam intensity”, and the “suprathreshold-density layer existing time v.s. beam intensity” can be thoroughly discussed.
  3. The Fig. 4(c) and Fig. 5(a) are both depicting the plasma density profile at t=-5 fs. However, the perturbation characteristics along x-direction at z/λs~ -5.3 and z/λs~ -3.5 in Fig. 4(c) cannot be clearly seen in Fig. 5(a).
  4. The resultant numerical spatial period of nanograting structure in this work, as shown in Fig. 5-7, is close to experimental data. How about the depth of those nanogratings along the z-axis? Comparison between numerical depth data and the existing experimental data would make this model more convincing.

Reviewer 2 Report

This paper presents the simulation of spatially periodic plasma formation inside silica glass with a focused ultra-short light pulse based on the Maxwell equations and the rate equation of the photoinduced plasma. In other theoretical studies of nanograting formation, it is assumed that silica glass has randomly distributed small inclusions which are responsible for the initial stage of spatially periodic plasma formation. On the other hand, the authors added noise ionization source which generates small fluctuations of the plasma density to the simulation and showed that the noise ionization sources causes the generation of spatially ordered plasma-field structure by irradiation with a single ultra-short light pulse.

To my best knowledge of laser-induced periodic structure formation inside silica glass, it is the first proposal that a noise ionization source should be responsible for the formation of the periodic structure. Their simulation showed clearly the formation of the periodic structure induced by the noise ionization source. However, I cannot understand what is the origin of a noise ionization source? Is there any experimental evidence on the noise? Randomly distributed small inclusions (nanobubbles or defects), which has been assumed the origin of the initialization of periodic plasma formation, are intrinsic for any glasses and even crystals. In addition, small inclusions can be accumulated in photoexcited regions after multiple laser shots. Actually, a number of experimental studies have shown that periodic nanostructures cannot be generated by a single laser irradiation. This fact suggests that small inclusions are reasonable sources for the formation of periodic structures rather than noise ionization source, and the simulation results in the paper that periodic plasma can be formed by a single laser pulse is inconsistent with the experimental studies. So, I think that only the simulation results are not enough evidence for the importance of noise ionization sources for the formation of periodic nanostructures in silica glass. In conclusion, this paper should not be accepted for the publication in the journal.

Other comments:

  1. A lot of sentences are too long, so it is difficult to understand what they mean. In addition, some sentences (lines 66-69 and lines 101-103) have too long subjects of three lines. So, it is better to ask several people to read some sentences.
  2. There is no explanation what are the basic equations to derive the equation (1).
  3. The author should explain that the amplitude of the noise amplitude source for the simulation is reasonable.
  4. The meanings of "breakdown pulse" and "breakdown wave front" are not clear.
  5. Please explain which colour in Figure 2 indicates what time in the simulation.

6. "the latter" in line 148 is not clear.

Reviewer 3 Report

The model presented relies heavily on the solution of Maxwell equations. The process of laser induced nanograting formation which is based on inhomogeneous energy absorption from the driving laser pulse cannot be oversimplified in this manner.

In my opinion the ionization and excitation of carriers in fused silica resulting in the formation of electron-hole plasma cannot be described by a simple rate equation for the used laser parameters. It is not very clearly explained how the plasma instability occurs.  Some preexisting inclusions are needed ("small foreign inclusions") and thus the model becomes even more unreliable if the goal is to show how the laser irradiation induces inhomogeneous energy absorption.

I also have doubts about the adequacy of the equation used for determining the energy deposition density in the medium under the irradiation conditions.

The authors talk about formation of surface plasmon within the material - isn't that just a bulk plasmon arising from the appearance of another component of the incident laser field due to laser induced modification of the medium through which the driving pulse is propagating (“internal surface plasmon”)? 

Citations are limited only to a quite narrow number of works by certain authors.